# Methods used to assess outcome consistency in clinical studies: A literature-based evaluation

Ewelina Rogozińska[1,2]*, Elizabeth Gargon[3], Rocío Olmedo-Requena[4,5,6], Amani Asour[2], Natalie A. M. Cooper[2], Claire L. Vale[1], Janneke van't Hooft[7]

1 Meta-Analysis Group, Institute of Clinical Trials and Methodology, MRC Clinical Trials Unit at UCL, London, England, United Kingdom, 2 Women's Health Research Unit, Queen Mary University of London, London, England, United Kingdom, 3 Department of Biostatistics, University of Liverpool, Liverpool, England, United Kingdom, 4 Department of Preventive Medicine and Public Health, School of Medicine, University of Granada, Granada, Spain, 5 Consortium for Biomedical Research in Epidemiology and Public Health (CIBERESP), Madrid, Spain, 6 Instituto de Investigación Biosanitaria ibs.GRANADA, Granada, Spain, 7 Meta-Research Innovation Center at Stanford (METRICS), Stanford University, Stanford, California, United States of America

* ewelina.rogozinska@ucl.ac.uk

**Data Availability Statement:** All relevant data are within the paper and its Supporting Information files.

## Abstract

Evaluation studies of outcomes used in clinical research and their consistency are appearing more frequently in the literature, as a key part of the core outcome set (COS) development. Current guidance suggests such evaluation studies should use systematic review methodology as their default. We aimed to examine the methods used. We searched the Core Outcome Measures in Effectiveness Trials (COMET) database (up to May 2019) supplementing it with additional resources. We included evaluation studies of outcome consistency in clinical studies across health subjects and used a subset of A MeaSurement Tool to Assess systematic Reviews (AMSTAR) 2 (items 1–9) to assess their methods. Of 93 included evaluation studies of outcome consistency (90 full reports, three summaries), 91% (85/93) reported performing literature searches in at least one bibliographic database, and 79% (73/93) was labelled as a "systematic review". The evaluations varied in terms of satisfying AMSTAR 2 criteria, such that 81/93 (87%) had implemented PICO in the research question, whereas only 5/93 (6%) had included the exclusions list. None of the evaluation studies explained how inconsistency of outcomes was detected, however, 80/90 (88%) concluded inconsistency in individual outcomes (66%, 55/90) or outcome domains (20%, 18/90). Methods used in evaluation studies of outcome consistency in clinical studies differed considerably. Despite frequent being labelled as a "systematic review", adoption of systematic review methodology is selective. While the impact on COS development is unknown, authors of these studies should refrain from labelling them as "systematic review" and focus on ensuring that the methods used to generate the different outcomes and outcome domains are reported transparently.

**Funding:** ER and CLV were supported by the UK Medical Research Council (MC_UU_12023/24).

**Competing interests:** The authors have declared that no competing interests exist.

## Introduction

Inconsistency (or heterogeneity) of outcomes measured in clinical studies is a widely recognised problem hindering evidence synthesis. [1–5] Core outcome sets (COS), defined as a minimum set of outcomes to be reported from all intervention trials sharing a common research objective, have been advocated as a solution to this problem. [6, 7] A growing number of studies aiming to develop a COS for conditions across a range of health areas [8, 9], is accompanied by reviews aiming to assess the consistency of outcomes in a formal way. [10–12]

Prior to the release of the Core Outcome Measures in Effectiveness Trials (COMET) handbook [13], guidance on a COS-related methodology was mostly focused on aspects of the consensus process. [14] Review of past research appears in the handbook in the context of assessing a need for a COS—described as an optional step—and informing a list of outcomes for a consensus process. The guidance on the conduct of this type of assessments is succinct and suggests the adoption of a systematic review approach. [13–15]

The method of systematically reviewing literature was introduced as a comprehensive way of summarising the evidence for the purpose of medical decision-making and identification of unanswered research questions. [16–19] The methodological rigour required for systematic review, which intends to minimise biases and provide a robust estimation of an underlying treatment effect, [16] requires considerable time and resources. [19, 20] The benefits of applying the same approach to the assessment of outcome consistency or generating a long list of outcomes for Delphi survey is unclear. [21]

Therefore, we set out to assess the methods adopted in evaluation studies of outcome consistency published to-date and examine their adoption of systematic review methods. Furthermore, we checked if identified evaluation studies were part of a COS project, examined methods specific to determining outcome consistency, such as type of collected outcomes, methods of their identification, and how authors assessed and presented outcome consistency or need for a COS.

## Materials and methods

Our work was guided by a prospectively developed protocol registered with PROSPERO (CRD42018100481).

### Identification of relevant studies

We included full texts of evaluation studies of outcome consistency in clinical studies on any health condition. We recognise that, even though, the COMET handbook refers to these type of evaluations as "systematic reviews" there is no consensus on the type of study design for this type of evaluation studies. Aiming to gain a thorough overview of the practices in this area and acknowledging lack of consensus regarding the study design, we decided to include any study design regardless of the design labelling. We searched the COMET database from its inception to June 2018; the search was updated in May 2019. [8] The COMET database is an annually updated repository of the international COS literature based on the systematic searches run in MEDLINE, SCOPUS, and Cochrane Methodology Register. [9] The search was supplemented with the resource of the Core Outcomes in Women's and Newborn Health (CROWN) initiative [10, 22] and check of references of the included studies. Systematic reviews of treatment effects with a secondary conclusion regarding outcome reporting, reviews of outcomes from non-clinical settings (registers, audits, population databases), COS protocols and reports mentioning outcome assessment but not describing them and studies focusing on a very narrow group of outcomes (e.g. only pain-related outcomes) were excluded.

## Data collection

All data were collected using a prospectively developed and piloted data collection form (S1 Text) and subsequently amalgamated into a master file in MS Excel. We collected information on publication year, medical speciality, evaluation's aims, scope (number of included studies and their type), whether the design was labelled as a "systematic review" and was it a part of a COS project (based on study acronym or the information provided in the publication). In order to examine the adoption of systematic review methodology, all evaluation studies (regardless of declared study design) were assessed against a tailored subset of A MeaSurement Tool to Assess systematic Reviews (AMSTAR) 2 items [23] we felt were relevant (S1 Table). AMSTAR was designed as a practical critical appraisal instrument enabling rapid and reproducible quality assessments of systematic reviews. In our work, we used items covering the development of a study protocol (items 1–3), identification and selection of eligible studies (items 4–5), data collection (item 6), reporting (item 7–8) and study quality assessment (item 9). The remaining items (10 to 16) were not assessed as we felt they were not applicable to evaluation studies of outcome consistency (e.g. study funding, aspects of meta-analysis, publication bias, etc.).

Information collected on the outcome-specific methods covered following elements: type of outcome (primary/secondary); a way of determining outcome type (clearly specified as primary or secondary, used in power calculation, etc.); approach to outcome extraction (any outcome reported in the publication or just those from the methods or results sections); use of any tool to assess the quality of outcome description; outcome unit as reported by the publication (individual outcomes or outcome groups); presentation of the findings (text, table, graphic format); and the conclusions. The conclusion section was examined for presence of any statement regarding inconsistency of assessed outcomes (heterogeneity, variation, etc.) or a need for a COS. Where detection of outcome inconsistency was concluded, we examined the methods and protocol of the evaluation, if available, for any description of how outcome inconsistency was defined. Where the need for a COS development was concluded, we looked for arguments given in the publications to support this conclusion. Double data extraction (AA, ROR or ER) and assessment of the subset of AMSTAR 2 items were performed on half of the included evaluations. Remaining evaluations were extracted and assessed by a single senior reviewer (ER). All discrepancies and statements supporting final judgements were recorded in the final Excel dataset.

## Strategy for data synthesis

Extracted data were cross-tabulated and presented either as frequencies or as median with corresponding interquartile range (IQR). When examining detection of outcome consistency, we assessed only studies with a full description of outcome assessment. Fisher's exact test was used to explore the potential difference between the evaluation studies labelled as "systematic review" and those that were not labelled as such. All descriptive summaries and graphs were computed with Excel software (MS Office 2016). The comparison of two groups—labelled and not labelled as "systematic review" was performed using Stata version 15.1.

## Results

### Study selection process and description of included studies

Out of 237 records assessed, 93 evaluation studies met the inclusion criteria—90 full reports and three summaries of outcome assessment in the final report of the COS project (S1 Fig). The main reason for records exclusion was that the objective was outside of this work's scope,

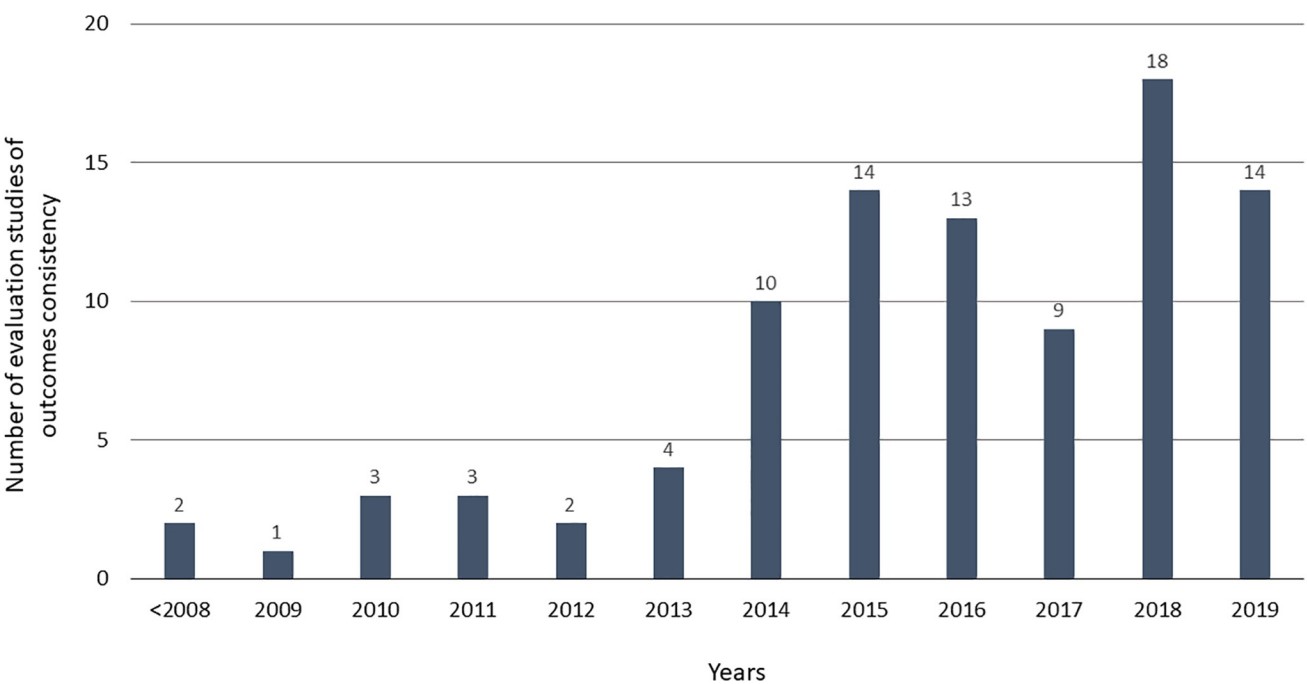

**Fig 1. Number of evaluation studies of outcome consistency in clinical studies over time.**

e.g. assessment of measurements, instruments, scales or definitions (S2 Table). Included evaluation studies were mainly published between 2014 and 2019 (Fig 1), included a median of 68 studies (Table 1) and covered topics across 24 medical areas (S3 Table). Around two-thirds (63/93) of the evaluation studies set out to assess outcome consistency or need for a COS in their objectives and around half (48/93) could have been linked to a COS project. Almost 80% (73/93) described their design as a "systematic review" (Table 1).

## Systematic review methodology in evaluations of outcome consistency

In most evaluation studies, we could identify a structured research question (87%, 81/93). Half of them declared having a protocol (50%, 46/93), of which 46% had been prospectively registered with PROSPERO (21/46) (S4 Table). In many, study identification (66%, 61/93) or data extraction (50%, 49/93) were carried out by two researchers (in duplicate). Authors rarely provided an exclusions list (5%, 5/93), a rationale behind the inclusion of specific study design (20%, 19/93), or performed quality assessment of included clinical studies (28%, 26/93) (Table 2). Even though in 91% of the evaluation studies (85/93) the literature search was performed in at least one bibliographical database (median of three databases per review) (S3 Table), we were able to classify only 33% (28/85) of them as comprehensive, as per AMSTAR 2 criteria, based on provided search details. The difference between evaluation studies labelled and not labelled as "systematic review" was statistically significant for the proportion of evaluation studies where study selection or data extraction were performed by at least two researchers, or the authors provided adequate description of included clinical studies (Table 2).

## Methods used to identify outcomes and their consistency

The number of identified outcomes was reported in 93% (86/93) of the evaluation studies with 40% (37/93) specifying from which section of the trial publication, the outcomes were

**Table 1. Characteristics of included evaluation studies of outcome consistency.**

| Characteristic | N | Descriptive |
|---|---|---|
| **Number of clinical studies per evaluation** *(median, Q1-3)* | 93 | 68 (34–133) |
| Distinction between primary and subsequent publications, n (%) | 93 | 30 (32) |
| **Objective(s) of evaluation study** | 93 | |
| To assess a need for COS or consistency in outcome selection, n (%) | | 63 (67) |
| To identify outcomes for Delphi survey in COS development, n (%) | | 18 (47) |
| To explore outcome-reporting, n (%) | | 13 (14) |
| **Evaluation study linked with a core outcome set project** | 93 | 48 (51) |
| **Evaluation study labelled as a "systematic review"** | 93 | |
| Yes, n (%) | | 73 (79) |
| No, n (%) | | 20 (21) |
| **When the evaluation study was not labelled as "systematic review", what other term was used to describe the design**: | 20 | |
| "Literature review" | | 11 (55) |
| "Review of outcomes" | | 2 (10) |
| "Analysis of outcome reporting" | | 1 (5) |
| "Electronic database search" | | 1 (5) |
| "Survey" | | 1 (5) |
| "Outcome mapping" | | 1 (5) |
| "Systematic exploration" | | 1 (5) |
| "Analysis of studies" | | 1 (5) |
| "Scoping review" | | 1 (5) |
| **Types of research included in the evaluation studies of outcome consistency** | | |
| **Primary research** | | |
| Full scale clinical study (e.g. RCTs, cohort studies), n (%) | 93 | 59 (63) |
| Feasibility or pilot study, n (%) | 93 | 8 (9) |
| **Secondary research (literature review)** | 93 | 27 (29) |
| **When the literature review was included, the purpose was to**: | | |
| Identify outcomes, n (%) | 27 | 10 (37) |
| Identify clinical studies for outcome assessment, n (%) | 27 | 15 (56) |

extracted. In two-thirds (63/93) of the evaluation studies, the authors made the distinction between primary and secondary outcomes, of which many also described how they ascertained whether the outcome was primary or not (41/63) (Table 3). Over half of the evaluation studies (55/90) looked at individual outcomes, 20% (18/90) assessed outcome domains (i.e. groups of individual outcomes referring to the same phenomena) and the remainder (19%, 17/90) applied both approaches (Table 3). Outcomes were frequently (90%, 81/90) presented in a tabulated format, and around one fifth (21%, 19/90) used a matrix to present outcome distribution (Fig 2).

We found statements on the inconsistency of outcomes in conclusions of 88% (80/90) of the assessed evaluation studies. None provided a description of how this inconsistency was defined or detected. The need for a COS was declared in 82% (74/90) and frequently justified by the encountered inconsistency of assessed outcomes (76%, 68/90) (Table 3).

## Discussion

The methods adopted in the evaluation studies of outcome consistency were variable, and none implemented all nine of the methodological expectations of systematic reviews as

**Table 2. Systematic review methods as specified in the subset of AMSTAR 2 items in evaluation studies of outcome consistency.**

| Item | Assessed aspect | Group | Yes (n,%)* | Unclear (n,%)* | No (n,%)* | N/A | Fisher exact test (p-value) |
|---|---|---|---|---|---|---|---|
| 1 | Did the research questions and inclusion criteria include the components of PICO? | Overall | 81 (87) | 9 (10) | 3 (3) | 0 | |
| | | Labelled as "Syst rev" | 64 (88) | 8 (11) | 1 (1) | | 0.129 |
| | | Not labelled as "Syst rev" | 17 (85) | 1 (5) | 2 (10) | | |
| 2 | Did the report contain an explicit statement that the methods were established prior to the conduct of the review and did the report justify any significant deviations from the protocol? | Overall | 46 (50) | 5 (5) | 42 (45) | 0 | |
| | | Labelled as "Syst rev" | 42 (58) | 4 (5) | 27 (37) | | 0.06 |
| | | Not labelled as "Syst rev" | 4 (20) | 1 (5) | 15 (75) | | |
| 3 | Did the authors explain their selection of the study designs for inclusion? | Overall | 19 (20) | 0 | 74 (80) | 0 | |
| | | Labelled as "Syst rev" | 17 (23) | 0 | 56 (77) | | 0.346 |
| | | Not labelled as "Syst rev" | 2 (10) | 0 | 18 (90) | | |
| 4 | Did the authors use a comprehensive literature search strategy? | Overall | 28 (33) | 4 (5) | 53 (62) | 8 | |
| | | Labelled as "Syst rev" | 23 (32) | 3 (4) | 43 (59) | | 0.200 |
| | | Not labelled as "Syst rev" | 5 (25) | 1 (5) | 10 (50) | | |
| 5 | Did the authors perform study selection in duplicate? | Overall | 61 (66) | 21 (23) | 10 (11) | 1 | |
| | | Labelled as "Syst rev" | 56 (77) | 11 (15) | 5 (7) | | <0.001 |
| | | Not labelled as "Syst rev" | 5 (25) | 10 (50) | 5 (25) | | |
| 6 | Did the authors perform data extraction in duplicate? | Overall | 49 (53) | 32 (34) | 12 (13) | 0 | |
| | | Labelled as "Syst rev" | 45 (62) | 21 (29) | 7 (10) | | 0.002 |
| | | Not labelled as "Syst rev" | 4 (20) | 11 (55) | 5 (25) | | |
| 7 | Did the authors provide a list of excluded studies and justify the exclusions? | Overall | 5 (6) | 0 | 84 (94) | 4 | |
| | | Labelled as "Syst rev" | 4 (5) | 0 | 66 (90) | | 1.00 |
| | | Not labelled as "Syst rev" | 1 (5) | 0 | 18 (90) | | |
| 8 | Did the authors describe the included studies in adequate detail? | Overall | 37 (42) | 2 (2) | 50 (56) | 4 | |
| | | Labelled as "Syst rev" | 35 (48) | 2 (3) | 33 (45) | | 0.006 |
| | | Not labelled as "Syst rev" | 2 (10) | 0 | 17 (85) | | |
| 9 | Did the authors assess the quality of included studies? | Overall | 26 (29) | 0 | 64 (71) | 3 | |
| | | Labelled as "Syst rev" | 22 (30) | 0 | 49 (67) | | 0.474 |
| | | Not labelled as "Syst rev" | 4 (20) | 0 | 15 (75) | | |

N/A, not applicable; Syst rev, systematic review;

*Percentages calculated using as denominator number without evaluation studies in N/A category;

**Table 3. Methods used to identify and assess outcomes in evaluation studies of outcome consistency.**

| Characteristic | n/N* (%) |
|---|---|
| **Evaluation studies reporting the number of identified outcomes (individual or domains)** | 86/90 (96) |
| Number of extracted outcomes (individual or domains) per review (median, Q1 to 3) | 80 (43 to 158) |
| **Distinction between primary and secondary outcomes** | 63/93 (68) |
| **When the distinction between the type of outcome(s) was made:** | |
| The primary outcome(s) was clearly specified as a primary | 27/63 (43) |
| The primary outcome(s) was clearly specified as a primary or used in the power calculation | 11/63 (17) |
| The primary outcome(s) was identified using other measures (e.g. outcome mentioned in the trial title, first reported) | 3/63 (5) |
| There were no details of how the primary outcome(s) was identified | 22/63 (35) |
| **Approach to outcome extraction** | |
| Outcomes mentioned anywhere in the report | 22/93 (24) |
| Outcomes mentioned only in methods | 11/93 (12) |
| Outcomes mentioned only in results | 4/93 (4) |
| Not specified or unclear | 56/93 (60) |
| **Quality assessment of outcome reporting** | 17/93 (18) |
| **Outcomes presented** | |
| Only individually | 55/90 (61) |
| Only grouped into domains** | 18/90 (20) |
| In both ways (individually and grouped in domains) | 17/90 (19) |
| **Presentation of extracted outcomes** | |
| Table of outcomes (individual or domains) with the number of studies | 81/90 (90) |
| Matrix (outcomes per study) | 20/90 (22) |
| Other graphic formats (e.g. bar chart, Venn diagram, Spiral graph) | 41/90 (46) |
| **Conclusion regarding outcome consistency** | |
| Detection of inconsistency in outcome choice | 80/90 (88) |
| Detection of a need for COS | 74/90 (82) |
| Due to identified inconsistency in outcome choice | 68/90 (76) |
| Due to other reasons than inconsistency in outcome choice | 6/90 (7) |
| **Conclusions of evaluations** | |
| Wide range of identified outcomes (individual or domains) | 55/90 (61) |
| Difference in how, rather than which, the outcome was measured | 52/90 (58) |
| Infrequent reporting of outcomes relevant to patient care | 37/90 (41) |
| Impact of inconsistency in outcome choice on evidence synthesis | 13/90 (14) |
| Difference in when the outcomes were measured | 12/90 (13) |

* Three reports (3/93) of full core outcome sets were excluded from some assessments (N = 90) as they focused only on providing a list of identified outcomes without reporting more details (see S3 Table)

** Outcome domain, constructs used to broadly classify individual outcomes referring to the same phenomena into a group

described in AMSTAR 2. The proportion of evaluation studies labelled as a "systematic review" and those that were not differed in three AMSTAR 2 items (duplicate identification of eligible studies, duplicate data extraction, and description of included clinical studies in sufficient detail). Most of the evaluations concluded that there was evidence of inconsistency in the outcomes used across clinical studies. Although none described how this inconsistency was determined, it was frequently cited to justify the need for COS development.

| Study ID | Outcome domain 1 | | | | Outcome domain 2 | | | | |
|---|---|---|---|---|---|---|---|---|---|
| | Outcome 1 | Outcome 2 | Outcome 3 | Outcome 4 | Outcome 5 | Outcome 6 | Outcome 7 | Outcome 8 | Outcome 9 |
| Study 1 | | ■ | ■ | | | | ■ | | |
| Study 2 | | ■ | | ■ | | | ■ | | ■ |
| Study 3 | ■ | ■ | | | | ■ | ■ | | |
| Study 4 | ■ | | | ■ | | | ■ | | |
| Study 5 | ■ | | ■ | ■ | | | | ■ | |
| Study 6 | | | ■ | | | | | | ■ |

**Fig 2. Matrix of outcomes—example.**

Our work provides the first extensive evaluation of methods used in evaluation studies of outcome consistency in clinical studies to date. It was guided by a prospectively developed protocol using broad inclusion criteria to facilitate identification of a representative sample of the studies of outcomes. Despite the benefits of publication of methodological study protocols are yet to be determined [23], their development can enhance the study's reproducibility. [24] Our methods heavily borrow from a systematic review process; however, our work is not a systematic review and should not be judged like one. For example, we did not set out to run an extensive search across numerous databases instead of relying on the COMET database and supplemented its results with the resources of the CROWN initiative. [25] Use of such a specific source of studies might be perceived as a limiting factor leading to the omission of some relevant publications. On the other hand, the COMET database is updated periodically, using extensive search strategy, for COS-related literature. [8] In this exploratory study of published literature, we aimed to obtain a pragmatic sample of studies using relevant sources. We did not search the international prospective register of systematic reviews (PROSPERO) [26], due to lack of specific indexing of this type of evaluation studies resulting in the inability to retrieve relevant records. As only less than a quarter of the included evaluation studies registered their

protocols with PROSPERO, we believe that omission of this source did not have a substantial impact on our work.

AMSTAR 2 was primarily developed to examine the robustness of systematic reviews of healthcare interventions. [27] Thus, the use of a subset of its items in this work might be contested. We decided to use AMSTAR 2 for two reasons, the COMET handbook reference to the evaluation studies of outcome consistency as "systematic reviews", and previous evaluation of COS-related literature in the area of women's health. [10] We selected the most relevant and applicable across a broader spectrum of study designs AMSTAR 2 items (S1 Table). Their application to all included evaluation studies allowed us to have a uniform comparison of the methods regardless of how they were labelled. Even though, a quarter of the evaluations in our sample did not self-identified as a "systematic review", the proportion of those satisfying AMSTAR 2 criteria labelled and not labelled as "systematic review" did not differ in six out of nine evaluated items.

Publication of the COMET Handbook [13] and related resources [14, 15, 28] increased access to guidance on robust methods for COS development. However, advice on methodology to evaluate outcomes to inform COS development is currently limited to a suggestion of adopting systematic review methodology. [13, 14] Evaluation studies of outcome consistency are a relatively new phenomenon with only a handful of this type of studies published before 2010 (Fig 1). Thus, the observed differences in the adopted methods probably arise from this limited guidance. In recent years, there has been a growing body of literature focusing on the methodological aspects of medical research [29]. The methodology of methods-orientated studies is inherently heterogeneous [29] and may involve features of a systematic review [21]. Nevertheless, the authors should refrain from labelling their work as "systematic review" when this is not the case [21], ensuring that their methodological explorations are guided by a robust methodology.

A well-executed systematic review is a resource-intensive and meticulous synthesis of available evidence. [16] Application of systematic review methods to assess outcome consistency raises a question over a value of applying such workload intense methodology—mainly in the form of researchers' time. Evaluation studies included in this work frequently aimed to assess the need for a COS and, simultaneously, collect outcomes to inform a subsequent Delphi survey. Delphi process tends to comprise multiple sources of information for generating the initial list of outcomes with a literature assessment of outcome being only one of them. Delphi enables involved stakeholders to propose relevant outcomes not yet captured on the list. [30] Thus, the rationale for conducting a systematic review to inform this process is unclear. Given that the impact of the methods of discussed evaluation studies of outcome consistency on the COS development process is unknown; authors should consider whether a systematic review approach is warranted.

None of the assessed evaluation studies specified how inconsistency of outcomes was detected; yet, the majority claimed in their conclusion to have identified such inconsistency. This finding highlights the need for more transparency in the design and reporting of these type of evaluations and is consistent with the conclusion of a recent COS literature. In their work, Young et al. examined number of reported outcomes, their definitions, timing, and approach to their grouping in 132 studies (development papers, protocols, and reviews) and found meaningful differences. [31] Based on their findings, they proposed a definition for a unique outcome and inconsistency of outcomes referred to in their work as outcome reporting heterogeneity (ORH). [31] The proposed definition brings attention to a difference between the number of identified outcomes (however defined) and their overlap between the clinical studies.

The methods of evaluation studies of outcome consistency published to date differ in their approach to the identification of eligible studies and their outcomes. Although frequently

labelled as "systematic review", their methods seldom satisfied the standards set for systematic reviews in the aspects of the process described by the first nine items of AMSTAR 2. In the absence of evidence of a benefit of adopting the systematic review approach in this context, we encourage researchers embarking on these type of project to follow general principles of systematic review practice. This should include the development of a research protocol with a clear description of intended methods and how researchers plan to determine if the main objective has been achieved and transparent reporting of the study. [32, 33] However, we discourage labelling this type of evaluation, as a "systematic review". Instead, we propose the use of terms such as "evaluation of outcomes" or "outcome mapping." Alternatively, authors may choose to use an established approach, such as "scoping review" and adopt the appropriate methodological and reporting standards [34, 35].

## Supporting information

**S1 Fig. Study selection flow diagram.**
(DOCX)

**S1 Text. Data collection form.**
(DOCX)

**S1 Table. Description of how A MeaSurement Tool to Assess systematic Reviews (AMSTAR 2) items were applied.**
(DOCX)

**S2 Table. List of assessed records with exclusion reasons.**
(XLSX)

**S3 Table. Detailed characteristic of included evaluation studies of outcome consistency.**
(DOCX)

**S4 Table. Details of identification of clinical study reports in the evaluation studies of outcome consistency.**
(DOCX)

## Acknowledgments

We would like to thank Professor Paula R. Williamson for her helpful comments at the initial stages of this work.

## Author Contributions

**Conceptualization:** Ewelina Rogozińska, Elizabeth Gargon, Rocío Olmedo-Requena, Natalie A. M. Cooper, Claire L. Vale, Janneke van't Hooft.

**Data curation:** Ewelina Rogozińska, Rocío Olmedo-Requena, Amani Asour.

**Formal analysis:** Ewelina Rogozińska.

**Investigation:** Rocío Olmedo-Requena.

**Methodology:** Ewelina Rogozińska, Elizabeth Gargon, Natalie A. M. Cooper, Claire L. Vale, Janneke van't Hooft.

**Project administration:** Ewelina Rogozińska.

**Supervision:** Ewelina Rogozińska.

**Writing – original draft:** Ewelina Rogozińska, Claire L. Vale.

**Writing – review & editing:** Ewelina Rogozińska, Elizabeth Gargon, Rocío Olmedo-Requena, Amani Asour, Natalie A. M. Cooper, Claire L. Vale, Janneke van't Hooft.

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
