## [Decision Letter · Decision Letter 0]

26 May 2020

PONE-D-20-08427

Evaluation of methods in studies of outcomes consistency in clinical studies

PLOS ONE

Dear Dr. Rogozińska,

Thank you for submitting your manuscript to PLOS ONE. After careful consideration, we feel that it has merit but does not fully meet PLOS ONE’s publication criteria as it currently stands. Therefore, we invite you to submit a revised version of the manuscript that addresses the points raised during the review process.

You can ignore the comment on highlighting  statistical significance (reviewer #2). 

We look forward to receiving your revised manuscript.

Kind regards,

Tim Mathes

Academic Editor

PLOS ONE

Reviewers' comments:

Reviewer's Responses to Questions

**Comments to the Author**

1. Is the manuscript technically sound, and do the data support the conclusions?

Reviewer #1: Yes

Reviewer #2: Yes

2. Has the statistical analysis been performed appropriately and rigorously? 

Reviewer #1: Yes

Reviewer #2: Yes

3. Have the authors made all data underlying the findings in their manuscript fully available?

Reviewer #1: Yes

Reviewer #2: Yes

4. Is the manuscript presented in an intelligible fashion and written in standard English?

Reviewer #1: Yes

Reviewer #2: Yes

5. Review Comments to the Author

Reviewer #1: The authors propose to evaluate the quality of the methods used in studies on outcomes consistencies. The topic is important and has not been investigated. However a few point could be improved

Major comments:

1) The author declare that their study was not a systematic review and reported having searched in the “COMET Database”. It is not clear what this COMET database is and the reference 8 does not mention it clearly either (the weblink of reference 8 is broken : http://www.comet-initiative.org/studies/search) It is time consuming for the reader to find out how this COMET database is provided (medline and scopus mainly? ).

ð In the method section, the authors should explain with a few sentences how the COMET database is fuelled.

2) The title of the article does not give any idea about the method they used. We propose to add more precision about the design of their study as a scoping review see the reference of the Joanna briggs institute (https://wiki.joannabriggs.org/display/MANUAL/Chapter+11%3A+Scoping+reviews) or of the PRISMA group (http://www.prisma-statement.org/Extensions/ScopingReviews)

ð “Evaluation of methods in studies of outcomes consistency in clinical studies: a scoping review”

3) At the end of the discussion, the author recommend not to use the term “systematic review” when studies are not : “However, we discourage labelling for this type of evaluation, as “systematic review” instead, proposing alternative terms such as an “evaluation of outcomes” or “outcome mapping”.”

ð We also suggest to use the term of “scoping review” in order to standardized the terminology according to PRISMA and the Joanna Briggs institute.

Minor comments:

Table 1 : we do not understand what “total number of participants per review” does mean. Who are these participants ? Is it the number of reviewers involved in each study or the number of patients included in the studies evaluated by the review? In case of the later, how is it relevant for the current study?

Reviewer #2: The abstract is not fully understandable without reading the main text. You should state concisely which type of studies are included in your review (e.g. synthesis studies evaluating outcome consistency in clinical studies). Please do not use undefined abbreviations (AMSTAR).

Please provide a brief description of AMSTAR in the main text.

Table 1: please move the notes into the table since they include relevant information

Table 2: please explain why you used Fisher's exact test (instead of chi-squared) and to which groups of studies you applied the test; please also highlight significant results (p<0.05).

Table 3: it is not understandable why denominators differ across the rows (90 or 93).

Tables (in general): when you use the word "studies", please verify whether it is easy to distinguish between studies included in your review and clinical studies included in the synthesis studies you examined

Please check all tables and figure captions and see if they can be more informative of the underlying content.

Please discuss any practical implications of your work (such as improvements in clinical research and practice).

Please identify any health area that requires more attention in future research.

6. PLOS authors have the option to publish the peer review history of their article (what does this mean?). If published, this will include your full peer review and any attached files.

Reviewer #1: No

Reviewer #2: No

---

## [Author Response · Author response to Decision Letter 0]

2 Jun 2020

Reviewer #1

1. The authors propose to evaluate the quality of the methods used in studies on outcomes consistencies. The topic is important and has not been investigated. However a few point could be improved 

We thank the reviewer for the recognition of the importance of this topic. 

2. The author declare that their study was not a systematic review and reported having searched in the “COMET Database”. It is not clear what this COMET database is and the reference 8 does not mention it clearly either (the weblink of reference 8 is broken: http://www.comet-initiative.org/studies/search) It is time consuming for the reader to find out how this COMET database is provided (medline and scopus mainly?). In the method section, the authors should explain with a few sentences how the COMET database is fuelled.

We agree with the reviewer that we had not adequately described the COMET database and have now addressed this shortcoming by adding the following description in the methods: 

 “We searched the COMET database from its inception to June 2018; the search was updated in May 2019. (8) The COMET database is an annually updated repository of the international COS literature based on the systematic searches run in MEDLINE, SCOPUS, and Cochrane Methodology Register. (9)

3. The title of the article does not give any idea about the method they used. We propose to add more precision about the design of their study as a scoping review see the reference of the Joanna Briggs Institute (https://wiki.joannabriggs.org/display/MANUAL/Chapter+11%3A+Scoping+reviews) or of the PRISMA group (http://www.prisma-statement.org/Extensions/ScopingReviews)

“Evaluation of methods in studies of outcomes consistency in clinical studies: a scoping review”

We thank the reviewer for the suggestion. However, our work was not set out to be a scoping review; thus we feel that labelling its design this way would be inappropriate. According to our protocol (see the PROSPERO registration) the design was described as “exploratory study” and was developed in the spirit of meta-research; discipline aiming to evaluate and improve research practices (Ioannidis et al. 2015 PLoS Biol. 13(10): e1002264). We have now added the information about the study design in the title as follows: “Methods to assess outcome consistency in clinical research: literature-baseded evaluation”

4. At the end of the discussion, the author recommend not to use the term “systematic review” when studies are not: “However, we discourage labelling for this type of evaluation, as “systematic review” instead, proposing alternative terms such as an “evaluation of outcomes” or “outcome mapping”.” We also suggest to use the term of “scoping review” in order to standardized the terminology according to PRISMA and the Joanna Briggs institute.

We agree that a scoping review may be another alternative and have amended the last sentence in our final paragraph as follows: 

“However, we discourage labelling for this type of evaluation, as “systematic review.” Instead, we propose terms such as an “evaluation of outcomes”, “outcome mapping.” Alternatively, authors ma choose to use an established approach, such as “scoping review”, and adopt the appropriate methodological and reporting standards (34, 35).”

5. Table 1: we do not understand what “total number of participants per review” does mean. Who are these participants? Is it the number of reviewers involved in each study or the number of patients included in the studies evaluated by the review? In case of the later, how is it relevant for the current study?

On reflection we agree with the reviewer and have now removed this information from Table 1.

Reviewer #2

1. The abstract is not fully understandable without reading the main text. You should state concisely which type of studies are included in your review (e.g. synthesis studies evaluating outcome consistency in clinical studies). Please do not use undefined abbreviations (AMSTAR).

We have now amended the abstract and checked the manuscript for consistency in the used terminology referring to included studies as “evaluations of outcome consistency in clinical studies”

AMSTAR abbreviation has been explained in the abstract as follows: “The methods were assessed using a subset of A MeaSurement Tool to Assess systematic Reviews (AMSTAR) 2 items (1-9) and additional outcome-specific questions.” 

2. Please provide a brief description of AMSTAR in the main text.

We have now added a following description in the methods section: “In order to examine adoption of systematic review methodology, all studies (regardless of declared study design) were assessed against a tailored subset of A MeaSurement Tool to Assess systematic Reviews (AMSTAR) 2 items (23) we felt were relevant to evaluated studies (S1 Table). AMSTAR was designed as a practical critical appraisal instrument enabling a rapid and reproducible quality assessments of systematic reviews.”

3. Table 1: please move the notes into the table since they include relevant information

The notes have been now incorporated into the main body of Table 1.

4. Table 3: it is not understandable why denominators differ across the rows (90 or 93).

The explanation of the difference is provided at the bottom of the table and has been rewritten for clarity: “*Three reports (3/93) of full core outcome sets were excluded from some assessments due to none applicability (for details see S3 Table)”. We also moved asterisks to the column heading (n/N*).

5. Tables (in general): when you use the word "studies", please verify whether it is easy to distinguish between studies included in your review and clinical studies included in the synthesis studies you examined

We have now amended all tables to improve their clarity according to the reviewer’s suggestion.

6. Please check all tables and figure captions and see if they can be more informative of the underlying content.

We have now checked all tables and figures to improve their clarity according to this suggestion.

7. Please discuss any practical implications of your work (such as improvements in clinical research and practice).

The practical implications of our work are discussed in the final paragraph of our discussion, notably that authors of these studies need to be transparent in reporting their methods and that they should not label such evaluations as systematic reviews if they do not fully adhere to stringent systematic review methodologies. 

Although we do not feel that our work has any direct impact on clinical practice; we hope that it will improve research practice relating to core outcome set development and reporting of such research.

8. Please identify any health area that requires more attention in future research.

As our work focuses on the methods rather than a specific clinical issue and is applicable across all health areas. We have recorded and reported the health areas in which evaluations of outcome consistency were carried out, however, we have not compared methodological practices between different health areas.

---

## [Decision Letter · Decision Letter 1]

5 Jun 2020

PONE-D-20-08427R1

Methods to assess outcome consistency in clinical research: a literature-based evaluation

PLOS ONE

Dear Dr. Rogozińska,

Thank you for submitting your manuscript to PLOS ONE. After careful consideration, we feel that it has merit but does not fully meet PLOS ONE’s publication criteria as it currently stands. Therefore, we invite you to submit a revised version of the manuscript that addresses the points raised during the review process.

We look forward to receiving your revised manuscript.

Kind regards,

Tim Mathes

Academic Editor

PLOS ONE

Reviewers' comments:

Reviewer's Responses to Questions

**Comments to the Author**

1. If the authors have adequately addressed your comments raised in a previous round of review and you feel that this manuscript is now acceptable for publication, you may indicate that here to bypass the “Comments to the Author” section, enter your conflict of interest statement in the “Confidential to Editor” section, and submit your "Accept" recommendation.

Reviewer #1: All comments have been addressed

Reviewer #2: (No Response)

2. Is the manuscript technically sound, and do the data support the conclusions?

Reviewer #1: Yes

Reviewer #2: Yes

3. Has the statistical analysis been performed appropriately and rigorously? 

Reviewer #1: Yes

Reviewer #2: N/A

4. Have the authors made all data underlying the findings in their manuscript fully available?

Reviewer #1: Yes

Reviewer #2: Yes

5. Is the manuscript presented in an intelligible fashion and written in standard English?

Reviewer #1: Yes

Reviewer #2: Yes

6. Review Comments to the Author

Reviewer #1: (No Response)

Reviewer #2: Thanks for addressing my previous comments. I have few additional minor suggestions:

1) replace "evaluations of outcome consistency" with "evaluation studies of outcome consistency"

2) if possibile, include such denomination among the keywords

3) check Table 1 caption (it should be outcome "consistency")

4) Table 3 caption: also in this case I would specify that methods are those of the included studies

5) Table 3: the third category ("when the distinction was made..") is a sub-category of the second one, and the way of reporting should reflect this

6) Table 3: the note is still unclear (you should fully understand the table without looking at supplementary files)

7) S3 Table caption: it should be "detailed characteristics" instead of "detail characteristics"

8) S4 Table caption is unclear: do you mean the clinical studies, or the evaluation studies that you assessed?

7. PLOS authors have the option to publish the peer review history of their article (what does this mean?). If published, this will include your full peer review and any attached files.

Reviewer #1: No

Reviewer #2: No

---

## [Author Response · Author response to Decision Letter 1]

14 Jun 2020

1 Thanks for addressing my previous comments. I have few additional minor suggestions:

We thank the reviewer for his further suggestions. We have reviewed all the points and amended the manuscript accordingly. 

2 replace "evaluations of outcome consistency" with "evaluation studies of outcome consistency" if possibile, include such denomination among the keywords

The term "evaluations of outcome consistency" has been changed to "evaluation studies of outcome consistency" and listed the term among the keywords.

3 check Table 1 caption (it should be outcome "consistency") 

The caption of Table 1 has been corrected to: “Characteristics of included evaluations of outcome consistency”

4 Table 3 caption: also in this case I would specify that methods are those of the included studies

The caption of Table 3 has been changed to: “Methods used to identify and assess outcomes in evaluation studies of outcome consistency”

5 Table 3: the third category ("when the distinction was made..") is a sub-category of the second one, and the way of reporting should reflect this 

The category and subcategories have been amended as follows:

When the distinction between the type of outcome(s) was made:

• The primary outcome(s) was clearly specified as a primary

• The primary outcome(s) was clearly specified as a primary or used in the power calculation

• The primary outcome(s) was identified using other measures (e.g. outcome mentioned in the trial title, first reported)

• There were no details of how the primary outcome(s) was identified

6 Table 3: the note is still unclear (you should fully understand the table without looking at supplementary files) 

The note has been amended as follows:

“Three reports (3/93) of full core outcome sets were excluded from some assessments (N = 90) as they focused only on providing a list of identified outcomes without reporting more details (see S3 Table)”

7 S3 Table caption: it should be "detailed characteristics" instead of "detail characteristics" 

The caption of S3 Table has been changed to: “Detailed characteristics of included evaluation studies of outcome consistency”

8) S4 Table caption is unclear: do you mean the clinical studies, or the evaluation studies that you assessed? 

The caption of S4 Table has been changed to: “Details of identification of clinical study reports in the evaluation studies of outcome consistency”

---

## [Editor Report · Decision Letter 2]

17 Jun 2020

Methods used to assess outcome consistency in clinical studies: a literature-based evaluation

PONE-D-20-08427R2

Dear Dr. Rogozińska,

We’re pleased to inform you that your manuscript has been judged scientifically suitable for publication and will be formally accepted for publication once it meets all outstanding technical requirements.

Kind regards,

Tim Mathes

Academic Editor

PLOS ONE
---

## [Editor Report · Acceptance letter]

22 Jun 2020

PONE-D-20-08427R2 

Methods used to assess outcome consistency in clinical studies: a literature-based evaluation 

Dear Dr. Rogozińska:

I'm pleased to inform you that your manuscript has been deemed suitable for publication in PLOS ONE. Congratulations! Your manuscript is now with our production department. 

Kind regards, 

on behalf of

Dr. Tim Mathes 

Academic Editor

PLOS ONE